# Equilibrium and thermodynamic studies of chromic overcrowded fluorenylidene-acridanes with modified fluorene moieties

Ya Wang [1], Yue Ma[1], Keisuke Ogumi[2,3], Bing Wang[1], Takafumi Nakagawa[4], Yao Fu[1] &
Yutaka Matsuo [1,2,4,5 ✉]

Chromic materials, an important class of stimuli-responsive materials, have aroused extensive attention in recent years. Normally, their color is based on changes in morphology. Few examples of chromic material based on conformational isomerization, such as in overcrowded alkenes, have been reported previously. Furthemore, experimental thermodynamic studies of overcrowded bistricyclic aromatic enes have not been carried out to our knowledge. Here, we show that *N*-phenyl-substituted fluorenylidene-acridanes, with a properly modified fluorene moiety, performs chromisms originating from conformational changes. Thermodynamic studies determine equilibrium constants, changes in enthalpy, entropy, and free energy in solution, enabling in-depth understanding of the equilibrium behavior of overcrowded alkenes and providing useful information for designing functional chromic compounds. Single-crystal X-ray diffraction analysis of fluorenylidene-acridanes in this work clearly shows well-tuned charge transfer from the acridane to the fluorene moiety. Various chromic behaviors such as mechanochromism, thermochromism, solvatochromism, vapochromism, and proton-induced chromism also support understanding of conformational isomerism.

[1] Hefei National Laboratory for Physical Sciences at the Microscale, School of Chemistry and Materials of Science, University of Science and Technology of China, 96 Jinzhai Road, 230026 Hefei, Anhui, China. [2] Department of Chemical System Engineering, Graduate School of Engineering, Nagoya University, Furo-cho, Chikusa-ku, Nagoya 464-8603, Japan. [3] Tokyo Metropolitan Industrial Technology Research Institute, 2-4-10 Aomi, Koto-ku, Tokyo 135-0064, Japan. [4] Department of Mechanical Engineering, School of Engineering, The University of Tokyo, 7-3-1 Hongo, Bunkyo-ku, Tokyo 113-8656, Japan. [5] Institute of Materials Innovation, Institutes of Innovation for Future Society, Nagoya University, Furo-cho, Chikusa-ku, Nagoya 464-8603, Japan. ✉email: yutaka. matsuo@chem.material.nagoya-u.ac.jp

The history of overcrowded bistricyclic aromatic enes (BAEs, Fig. 1) can be traced back almost one-hundred years, when they were used to discover the phenomenon of thermochromism[1]. BAEs have a general structure consisting of two tricyclic moieties connected with a central double bond (Fig. 1a)[2,3]. The overcrowding in BAEs causes out-of-plane deformations to alleviate unfavorable steric interactions between the nonbonded atoms in the fjord regions. BAEs have been attractive systems for materials scientists especially since system such as bianthrone and dixanthylene have been found to exhibit thermochromism, piezochromism, and photochromism[4,5].

Chromic molecules with ability to undergo color changes in response to external stimuli such as heat, pressure, radiation, and fuming have attracted great attention as smart materials with numerous potential applications in sensors, memory, and optical materials[6–9]. A variety of systems have been developed by utilizing the chromic phenomena induced by breaking or forming bonds, changing packing structures, and controlling conformational change[10–15]. However, to meet the requirements for practical applications, the chromic materials should easily revert back to their initial state from the perturbed state. In this regard, chromic materials with conformational change are believed to have better functionality in the control of various efficient and reversible chromic processes. With improved functionality based on conformational chromism, potential applications have been suggested in force-responsive smart materials, data storage, green printing, and anticounterfeit labels.

Herein, we describe the synthesis of fluorenylidene-acridanes (FAs) with functionalization of the fluorene moiety (denoted as F'A) to introduce perturbations more directly into the π-conjugation systems of FAs, and we demonstrate various chromic

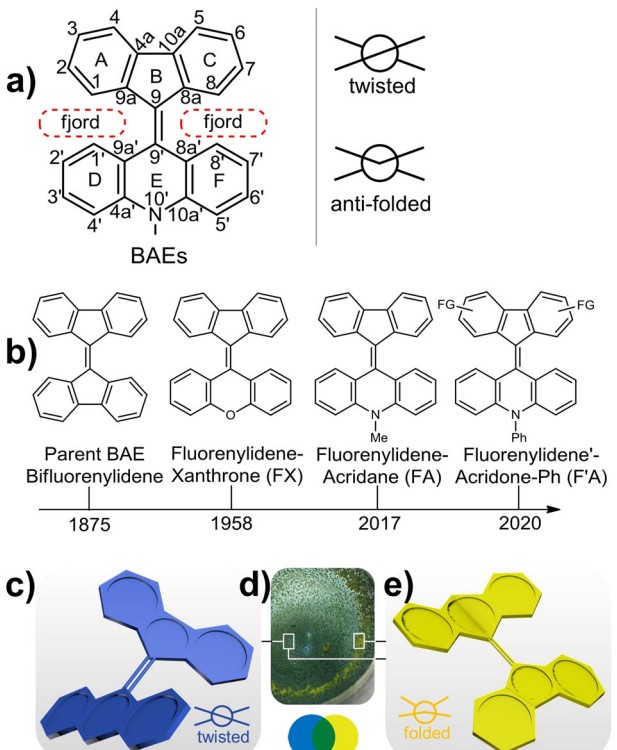

**Fig. 1 Overcrowded bistricyclic aromatic enes. a** Structure and atom labeling of FAs (fluorenylidene-acridanes), **b** overview of BAEs (bistricyclic aromatic enes), **c, e** schematic drawings for twisted (blue) and folded (yellow) conformers of Ph-F'As (**5a**), respectively. **d** Photograph of a mixture of the two conformers, appearing green due to mixing of the blue and yellow conformers.

behaviors in the ground and excited states in greater detail than in our previous reports[16,17]. The equilibrium between two conformers of BAE molecules has long been discussed, but the equilibrium and thermodynamic behaviors have seldom been explored in detail. Most importantly, in this work, we discuss equilibrium constants in terms of ratios of two conformers at different temperatures to investigate changes in enthalpy, entropy, and Gibbs free energy arising from this conformational isomerism.

## Results

**Design, synthesis, and characterization.** Small energy difference between the two conformers of BAEs have been experimentally known[18] (Supplementary Figs. 72, 73, Supplementary Table 22, Supplementary Note 4, 5, Supplementary Data 1–4). Our calculations for FAs with a M06/6–31G(d) method showed that the energy differences between the twisted and folded conformations are very small and that the folded conformer was slightly favored. The energy differences between the two conformers for **5e** was calculated as only 0.86 kJ mol$^{-1}$, indicating that neither one is overwhelmingly favored. Considering that modification of the phenyl group on our previous Ph-FAs[17], we envisioned that conformational isomerism could be further controlled by modifying the electron-accepting fluorene moiety with electronic-altering and steric-altering functional groups. Direct substitution on the FA cores could largely change the stability and crystallinity of the conformers.

To realize the advantages of FAs over fluorenylidene-xanthenes[2,3], FA derivatives with various substituents on the nitrogen atom of the acridane part are possible[7]. In this work, to gain further insights into the factors governing the ratio of conformers for approaching the shift of the equilibrium toward the folded confirmer for chromic studies, a new synthesis strategy was employed in which several functional groups were introduced onto the fluorene moiety of Ph-F'As. Stability of the folded conformer is essential for chromism. We anticipated that an electron-donating substituent on fluorene would suppress charge transfer from the electron-donating acridane moiety to the electron-withdrawing fluorene moiety, thus stabilizing the folded conformer. To implement this synthetic strategy in practice, the functional groups were installed onto the fluorene moiety prior to Barton–Kellogg cross-coupling rather than modifying the Ph-FAs directly (Fig. 2). Ph-F'As were generally synthesized by the Barton–Kellogg olefination from thioacridones and diazofluorenes[19,20], both of which were modified in advance. A moderate electron-withdrawing group, a phenyl group, was introduced onto the nitrogen atom to decrease the negative charge distribution in the C=S bond and accelerate nucleophilic attack by the diazo compounds in this 1,3-dipolar cycloaddition. Copper-catalyzed Ullmann amination was used to replace the hydrogen atom on the nitrogen atom with the phenyl group, followed by sulfuration using Lawesson reagent to form thioketones. Several synthetic routes were separately used to prepare various fluorenones. Pd-catalyzed oxidative intramolecular C–C coupling was used to prepare **1a** and **1b**, while alkylation of the phenolic hydroxy group with dimethyl sulfate was performed to obtain **1c**. Compound **1d** is commercially available. Nitration and Suzuki coupling were conducted to synthesize **1e** and **1f**, respectively. After that, fluorene derivatives **1a–f** were reacted with hydrazine to form hydrazones (**2a–f**), which were then oxidized to afford various diazo compounds (**3a–f**). Desulfurization of the episulfide was accomplished using PPh$_3$ in one-pot in the Barton–Kellogg reaction, without isolating **4**[17].

Ph-F'As were characterized by single-crystal X-ray diffraction (XRD), $^1$H nuclear magnetic resonance (NMR), $^{13}$C NMR, high-resolution mass spectrometry, UV–Vis absorption spectroscopy, and fluorescence spectroscopy (for details, see Supplementary

**Fig. 2 Synthetic routes to Ph-F'As with various substituents on the fluorene moiety.** Inset: photographs of Ph-F'A crystals or powder (**5a–f**). FG functional groups.

Figs. 9–48, 53, 54). Note that [1]H NMR spectra exhibit one major peaks set of one isomer and another minor peaks set of the other isomer[2,21–25]. Except for **5f**, all other Ph-F'As **5a–e** had exceptionally high crystallinity. The twisted conformers with partial zwitterionic character were stabilized by polar solvent molecules, such as dichloromethane (DCM) and acetonitrile, and through dipole–dipole interaction (Supplementary Fig. 51). Crystal structures of **5a**, **5b**, **5c**, **5d**, and **5e** were determined by single-crystal XRD. Both folded and twisted conformers were identified for **5a** and **5e**, which is rare in the history of BAE structural determination[5,7]. To our surprise, a **5a** crystal containing both the folded and twisted conformers in one single unit cell was obtained.

**Equilibrium study based on variable-temperature UV–Vis spectra.** As we proposed in our previous work[17], the absorption band around 430 nm involves absorption of both folded and twisted conformers. To analyze each conformer's contribution to the absorption, we performed variable-temperature UV–Vis spectroscopy (Supplementary Fig. 52). In Fig. 3a–d, we can see that as the temperature was increased from 30 to 110 °C, the absorption band around 680 nm decreased due to the decrease in the twisted conformer, whereas both increases and decreases in absorption were observed in the short-wavelength region. The latter observation confirms that both of the conformers contributed to absorption in this region. Interestingly, as the temperature was increased, the equilibrium shifted from the twisted conformer to the folded one.

Based on La Chatelier's principle, this would indicate that the conformational change from twisted to folded was endothermic in this equilibrium. We checked the reversibility of the spectral changes after cooling the solutions from 110 to 30 °C and found that the spectra were the same as before heating.

The variable-temperature UV–Vis spectra for **5e** was more clearly interpreted than those for **5a–d**, because of less overlapping absorption at the short-wavelength region. With increasing temperature, the spectra of **5e** exhibited three decreases in peak intensity (around 680, 360, and 310 nm) and one increase in peak intensity (around 430 nm). Based on solid UV–Vis absorption spectra, the color of twisted single crystals, and DFT calculation results for **5e** (Supplementary Fig. 72), we assigned the absorption band around 680 nm to the twisted isomer. Thus, according to the law of conservation of mass, the three peaks around 310, 360, and 430 nm were assigned to twisted, twisted, and folded conformers, respectively.

Overlapping absorption of two conformers normally makes it difficult to calculate the ratio of two isomers at equilibrium. Fortunately, in this work, absorption of the folded conformer of **5e** predominates in the absorption around 430 nm. Using a Gaussian fitting method in matlab-2019b for peaks deconvolution, we determined the percentage of absorption by folded **5e** at 430 nm to be 92.5% (Fig. 3f). Thus, the equilibrium between conformers of **5e** was studied based on UV–Vis absorbance Eq. (1).

Based on the law of conservation of mass and the Beer–Lambert Law, solving the following simultaneous equations can provide the ratio of the folded conformer to the twisted one.

$$\mathbf{5e}_{\text{twisted}} \rightleftharpoons \mathbf{5e}_{\text{folded}} \qquad (1)$$

$$\Delta c_{\text{folded}} + \Delta c_{\text{twisted}} = 0 \qquad (2)$$

$$A = \varepsilon \cdot c \cdot L \qquad (3)$$

Here, $A$, $\varepsilon$, $c$, and $L$ are the absorbance, molar absorption coefficient, molar concentration, and path length of the absorption layer, respectively. Thus, we can obtain the following equations:

$$\left(A_{\text{folded}}^{T1} - A_{\text{folded}}^{Tn}\right)/\varepsilon_{\text{folded}} + \left(A_{\text{twisted}}^{T1} - A_{\text{twisted}}^{Tn}\right)/\varepsilon_{\text{twisted}} = 0. \qquad (4)$$

$$c_{\text{folded}}^{Tn}/c_{\text{twisted}}^{Tn} = \left(A_{\text{folded}}^{Tn}/\varepsilon_{\text{folded}}^{Tn}\right)/\left(A_{\text{twisted}}^{Tn}/\varepsilon_{\text{twisted}}^{Tn}\right). \qquad (5)$$

$A_{\text{folded}}^{Tn}$ and $A_{\text{twisted}}^{Tn}$ are the absorbance of the folded and twisted conformers at temperature $n$ ($T_n$, $n = 1, 2\ldots6$), respectively. In this analysis, $T_1$, $T_2$, $T_3$, $T_4$, $T_5$, and $T_6$ are 30 °C, 40 °C,

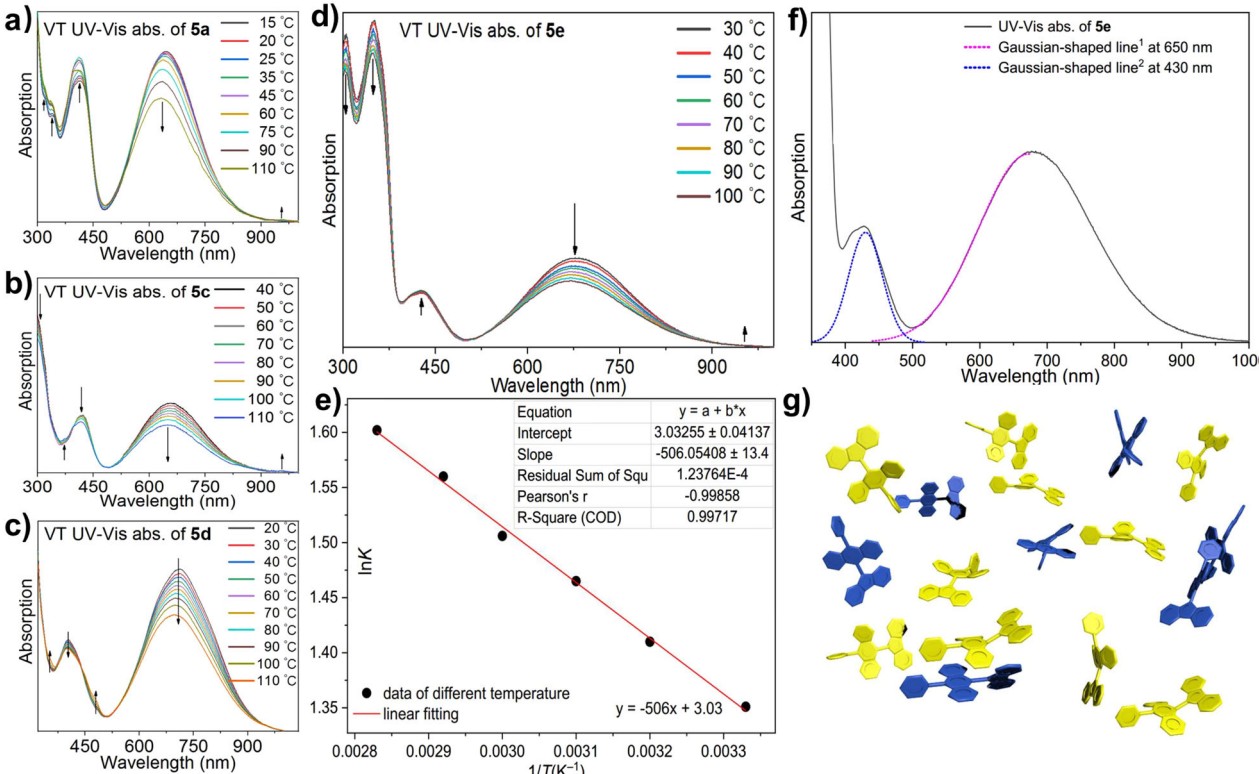

**Fig. 3 Variable-temperature UV–Vis spectra and analysis.** Spectra of **a 5a**, **b 5b**, **c 5c**, **d 5e** in DMF, and **e** Van't Hoff plot ($1/T$ vs. ln$K$) with a linear fit at 30, 40, 50, 60, 70, and 80 °C, **f** Gaussian fitting of UV–Vis absorption of **5e**, **g** Schematic drawing for equilibrium between folded and twisted aryl-F'As. Gaussian-shaped line[1] and line[2] in **f** was fitted as $y = e^{(-0.9085 \times (x/100)^2 + 12.2173 \times (x/100) - 41.4515)}$ and $y = e^{(-7.8904 \times (x/100)^2 + 67.8576 \times (x/100) - 146.8101)}$, respectively.

50 °C, 60 °C, 70 °C, and 80 °C, respectively. Molar absorption coefficients of the folded and twisted conformers in $T_n$ are denoted by $\varepsilon_{folded}^{Tn}$ and $\varepsilon_{twisted}^{Tn}$, respectively. The concentrations of the folded and twisted conformers at $T_n$ are denoted by $c_{folded}^{Tn}$ and $c_{twisted}^{Tn}$, respectively. From the absorption spectra in Fig. 3d, the ratio of molar absorption coefficients between the folded and twisted conformers $\varepsilon_{folded}/\varepsilon_{twisted}$ was calculated as 1/6.84 ($T_n = 80$ °C, Supplementary Figs. 77–78, Supplementary Note 6). Thus, from Eq. (5) and the contribution of the folded conformer to the absorbance at 430 nm, the ratio $c_{folded}/c_{twisted}$ at 30 °C (i.e., the equilibrium constant, $K$) was calculated to be 3.86.

We found the above-mentioned equilibrium constant ($c_{folded}/c_{twisted} = K = 3.86$ at 30 °C) surprising because it meant that the concentration of the folded conformer was greater than that of the twisted one at equilibrium, but **5e** in solution normally exhibited the dark green color of the twisted conformer. This was due its large molar absorption coefficient of the twisted conformer. To further investigate this equilibrium, we created a Van't Hoff plot ($1/T$ vs. ln$K$; Fig. 3e) to determine the changes in enthalpy, entropy, and Gibbs free energy ($\Delta H$, $\Delta S$, and $\Delta G$, respectively). From

$$\Delta G = \Delta H - T \cdot \Delta S. \qquad (6)$$

$$\Delta G = -R \cdot T \cdot \ln K. \qquad (7)$$

It can be obtained that the Van't Hoff equation:

$$\ln K = -\Delta H^o/(R \cdot T) + \Delta S^o/R \qquad (8)$$

Here, $K$ is the equilibrium constant (conversion from twisted to folded, $K = c_{folded}/c_{twisted}$), $T$ is the absolute temperature, and $R$ is the universal gas constant. Plotting ln$K$ vs. $1/T$ allows the determination of $\Delta H^o$ and $\Delta S^o$, where $\Delta H^o$ and $\Delta S^o$ are the

standard changes in enthalpy and entropy, respectively. From Fig. 3e, we see a good linear relationship between the two variables ($R^2 = 0.997$). $\Delta H$ was calculated to be 4.21 kJ mol$^{-1}$ (Supplementary Table 23). This result is in accordance with our assumption based on La Chatelier's principle with the positive value of $\Delta H$ indicating the transformation from the twisted to folded conformer is an endothermic process. $\Delta G$ was calculated to be –3.43 kJ mol$^{-1}$, which means the folded conformer is a more stable form. This agrees with the equilibrium constant we determined. As expected, the calculated change in entropy was very small (0.0252 kJ mol$^{-1}$), because of an intramolecular transformation involving only conformation isomerism. We note that this is the first time that thermodynamic parameters ($\Delta H$, $\Delta S$, and $\Delta G$) have been determined for two BAE conformers of in equilibrium.

**Chromisms in the ground state and excited state.** Chromism based on conformational change is very rare among mechanochromic materials, which normally occurs due to morphological differences altering intermolecular interaction[12,13]. The folded conformer is preferred especially for **5a**, **5c**, and **5e** in the solid state, enabling various kinds of chromism. Intuitive experiments involving mechanochromism, thermochromism, solvatochromism, vapochromism, and proton-induced chromism (Fig. 4 and Supplementary Figs. 63–70) were designed to observe the conformational isomerism. For chromism induced by mechanical stimuli (Fig. 4a), we used an iron rod to draw on a paper absorbed with folded **5a**, which obtained after heating a twisted-absorbed paper in 60 °C to induce conformational isomerization. Vapor-induced chromism is shown in Fig. 4b. Putting the green drawings in a closed vessel filled with DCM vapor at 25 °C, the drawings gradually disappeared within 3 min. We surmise this

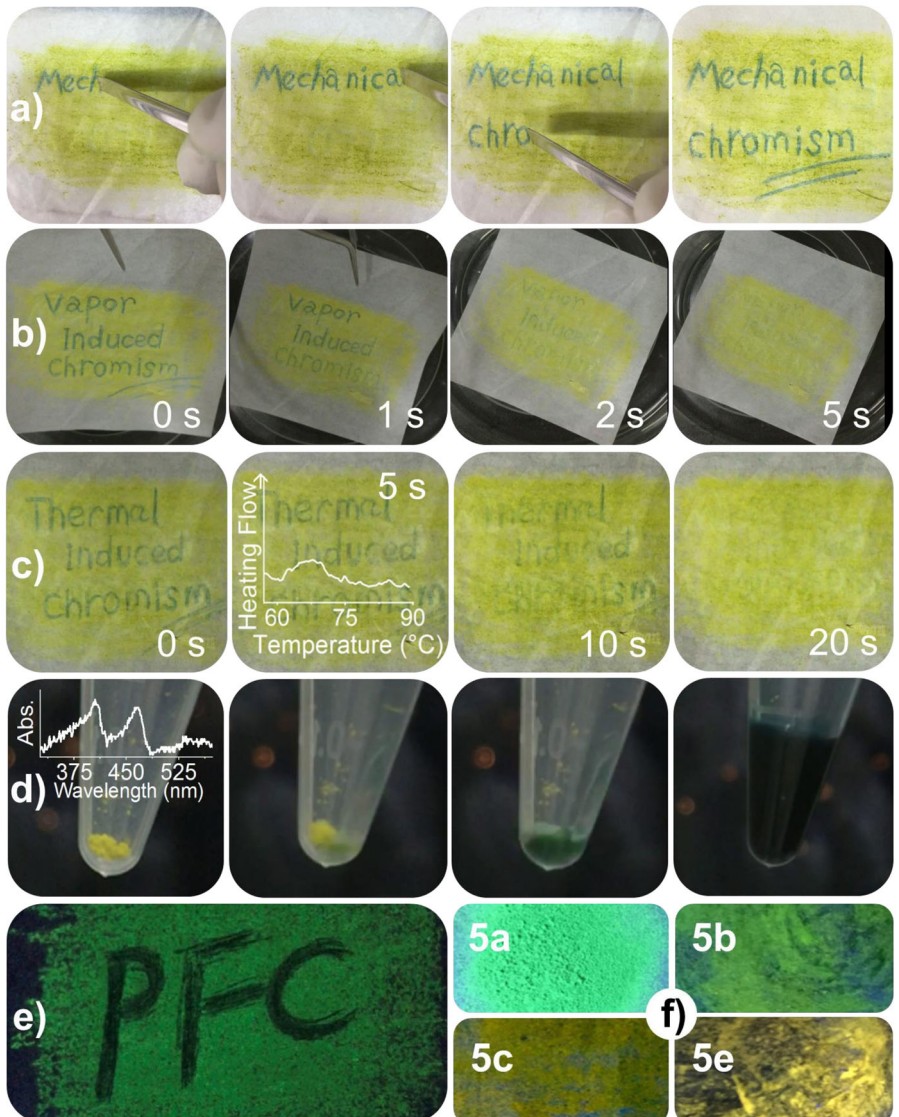

**Fig. 4 Various chromic behaviors and fluorescence properties. a** Mechanochromism, **b** vapochromism from green to yellow in response to DCM vapor, **c** thermochromism at 70 °C; overlaid graph: DSC curve, **d** dissolution-induced color change with DCM; overlaid graph: solid-state UV spectrum, **e** piezofluorochromism of **5a** after drawing under 365 nm excitation, **f** photographs of photoluminescence for **5a** (powder), **5b**, **5c**, **5e** (absorbed in paper).

color change was due to microcrystallization induced by the solvent vapor. Color change from green to yellow was also triggered by thermal-induced chromism (Fig. 4c). When the drawings were placed on a hot plate equipped with a thermocouple, the drawings disappeared within 20 s at 70 °C. We examined the robustness of repeated drawing and erasing and confirmed repeatability for at least 50 cycles. After this repeated drawing/erasing test, $^1$H NMR measurements showed no evidence of undesired reactions. Decomposition temperature of **5a**, **5b**, **5c**, **5d**, and **5e** were 216 °C, 243 °C, 255 °C, 315 °C, and 261 °C, respectively (Supplementary Fig. 49). These results suggest the potential utilization of these characteristics for environmental-friendly printing materials, in which a printer can print by applying pressure without ink and drawings can be erased by heat. This behavior can be explained by recrystallization to the folder conformer over the glass transition temperature (around 70 °C). Differential scanning calorimetry (DSC) measurement of **5a** supports this, showing a broad and small exothermic peak around this temperature. For other Ph-F'As, this exothermic peak ranged from 65 to 75 °C (Supplementary Fig. 50). Transformation

from the twisted to folded conformers in the solution state was endothermic, which was certified by the positive value of $\Delta H$. On the other hand, thermal ordering from the twisted to the folded forms in the solid state occurs by heating around at 70 °C with exothermic process because of intermolecular interaction such as packing force in the folded crystal, like fuming-induced crystallization into the folded one. In Fig. 4d, a bluish green solution formed after addition of DCM to the yellow powder. This color change was caused by formation of the equilibrium between the folded and twisted conformers in DCM solution. Supplementary Fig. 79a shows the color change from yellow to green due to melting. Melting was observed after putting the paper on a hot plate around 210 °C. This was consistent with DSC data showing an endothermic peak around 210 °C.

The Ph-F'As also exhibited excellent piezofluorochromism (PFC). By introducing a series of functional groups onto the fluorene moiety, the photoluminescence spectra bands were tuned over a wide range of 60 nm (Supplementary Fig. 80a). The corresponding photoluminescence of **5a–c** and **5e** under 365 nm excitation is shown in Fig. 4f, inset. Extremely weak luminescence

was observed after grinding even with application of little force (Fig. 4e, inset). This phenomenon can be explained by the conformational change from the luminescent folded conformer to the non-luminescent twisted conformer by grinding and energy transfer from the former to the latter. Initially, there were aggregates of the folded conformer, but grinding formed a small amount of the twisted conformer that dropped out of the folded aggregates. The twisted conformer had a smaller energy gap than the folded conformer and was nonluminescent due to partial charge transfer. Consequently, the small amount of the twisted conformer acted as a luminescence quencher in PFC. Differences in fluorescence due to conformational isomerism have been also reported in some Au(I) complexes and in a one-dimensional Cu (I) coordination polymer[26,27]. With these special properties, Ph-F'As have the potential to serve as excellent organic PFC materials[8,28,29].

**Substituents and solvents effect for charge separation and reversible protonation.** UV–Vis absorption spectroscopy was performed to explore the effect of substituents on the equilibrium constant between the folded and twisted conformers of **5a–f** in DCM solution (Fig. 5a). With six kinds of functional groups introduced onto the fluorene moiety, the absorption peak for the charge transfer band was tuned over a wide range from 620 to 750 nm, which was assigned to the twisted conformers. Compared with **5a** and **5c**, a red-shift was observed for **5f**. This red shift can be attributed to increased charge transfer from the acridane moiety to the fluorene moiety because of the strongly electron-withdrawing nitro group. Compounds **5a** and **5c** have two electron-donating groups at the 3,6-position and 2,7-position of the fluorene moiety, respectively, and their charge transfer bands are shifted to shorter wavelengths (630–640 nm). The UV–Vis absorptions of **5a–f** in the range 300–400 nm were quite

different from one another, reflecting the involvement of the substituents on fluorene in the absorption. For example, **5e** with thienyl groups and **5f** with nitro groups shows distinctive absorption around 300–350 nm due to the $n \rightarrow \pi^*$ transition involving the lone pairs of the S and O atoms, respectively.

Nine solvents, with a wide range of solvent polarity parameter values ($E_T(30)$ from 31.0 to 55.4) were selected to analyze the relationship between solvent polarity and the absorption ratio of the folded and twisted conformers. Positive correlations were observed between $E_T(30)$[30] and the ratios of absorption around 670 nm and around 430 nm (Fig. 5b, c). This indicates that the equilibrium shifted from the folded conformer toward the twisted conformer with increasing solvent polarity[31]. In addition, the solvent effect also manifested as a red-shift with increasing solvent polarity[32]. Methanol was a notable exception to the above trend, which was attributed to the effect of aggregation, as described in our previous work[17], and probably also to the effect of equilibrium between FAs and methanol adducts. Both increasing intensity and red-shift of absorption band for the twisted conformer suggest conformational change-enhanced solvatochromism. Because of this boosted color change, Ph-F'As have the potential to be a new kind of solvatochromic dye[33].

As seen from Fig. 5d, it is easy to conclude that **5a** was protonated with increasing acidity of the solution. The partially zwitterionic twisted Ph-F'As conformer would form by deformation from the overcrowded structure accompanying charge transfer from the acridane moiety to the fluorene moiety (Fig. 6a). When Ph-F'As molecules were surrounded by polar acetic acid, intramolecular charge transfer was further promoted, leading to protonation of the fluorene group to form bright yellow protonated **5a**. This process was reversed when an organic base like triethylamine was added (Supplementary Fig. 79c).

Electrophilic properties were also explored via addition of methanol to the central double bond of **5a**. Formation of the

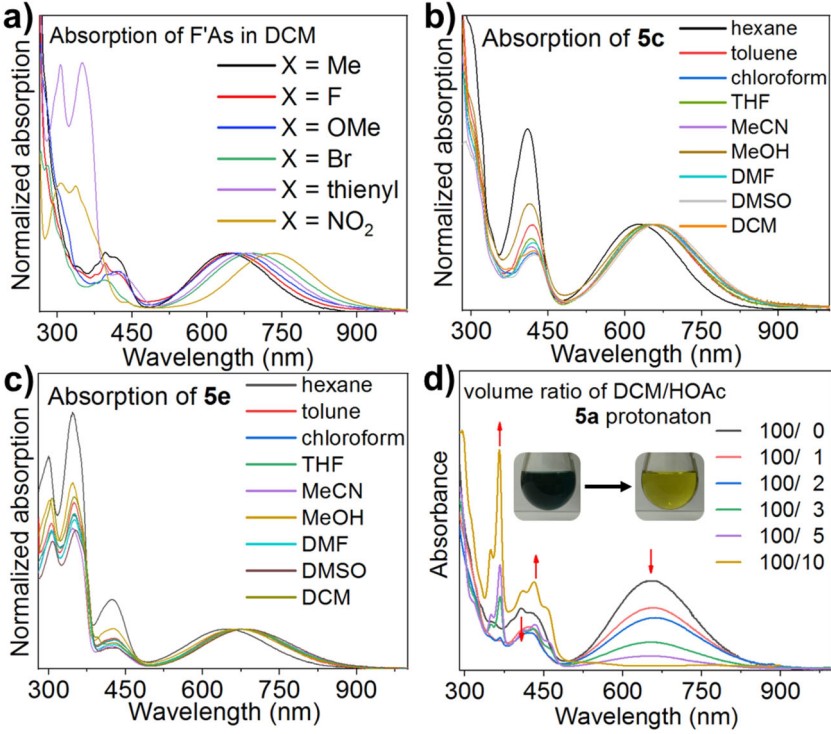

**Fig. 5 UV–Vis absorption spectra. a** Normalized absorption spectra of Ph-F'As in DCM, normalized absorption spectra of **b 5c** and **c 5e** in different solvents, **d** absorption spectra of **5c** at the same concentration in different ratios of DCM/glacial acetic acid. Inset: **5a** in different ratio of HOAc/DCM mixture.

**Fig. 6 Protonation and electrophilic reaction of 5a. a** Nucleophilic properties of **5a** and mechanism for protonation of **5a** by acetic acid, **b** electrophilic properties of **5a** and mechanism for addition of methanol to **5a**.

methanol adduct **6a** proceeded as shown in Fig. 6b. Considering pKa of methanol is 15.5, we assume the reaction occurs via a concerted mechanism. The reaction started with attack by a lone pair of methanol on the slightly cationic acridane moiety, and then the fluorene moiety is protonated. The X-ray structure of **6a** is shown in Fig. 7. Notably, the methanol adduct could easily undergoes elimination to reform **5a** unless it was in a crystalline state.

**Electrochemistry and charge carrier transport.** Cyclic voltammograms of **5a**–**e** in DCM (vs. Fc/Fc$^+$) showed reversible one-electron oxidation and one-electron reduction (Supplementary Fig. 71). The first oxidation potentials ranged from –0.24 to –0.10 V vs. Fc/Fc$^+$, which are comparable to our previous data for Me-FA and Ph-FA (–0.17 V and –0.19 V, respectively)[16,17]. Compound **5a** exhibited the lowest oxidation potential with a negative value (–0.24 V), showing facile oxidation compared with ferrocene because of the electron-donating methyl group on the fluorene moiety. On the other hand, **5d** showed relatively a high oxidation potential of –0.10 V vs. Fc/Fc$^+$ due to the electron-withdrawing bromo group on the fluorene moiety. The reduction potentials of **5a**–**e** ranged from –1.59 to –1.86 V vs. Fc/Fc$^+$. Among the five compounds tested, smaller bandgaps were observed in **5d** (1.69 V) and **5e** (1.87 V), which had bromo and thienyl groups, respectively. The largest bandgap was observed in **5a** (2.08 V). In addition, a very small reduction wave around –0.9 V was observed for **5a**–**e** and was assigned to reduction of the folded conformers[30,34–36].

We measured the charge carrier transport properties of Ph-F'As by the space-charge limited current (SCLC) method[37] (Supplementary Figs. 74–76). Hole and electron devices were fabricated with configurations of ITO/PEDOT:PSS/Ph-F'As/MoO$_3$/Al and ITIO/ZnO/Ph-F'As/Al, respectively. The hole mobilities of **5a**, **5c**, **5d**, and **5e** were $6.6 \times 10^{-7}$, $1.1 \times 10^{-7}$, $1.1 \times 10^{-6}$, and $6.3 \times 10^{-7}$ cm$^2$ V$^{-1}$ s$^{-1}$, respectively. Measurement with **5b** failed due to its high crystallinity. The electron mobilities of **5a**, **5c**, and **5d** were $2.5 \times 10^{-8}$, $1.4 \times 10^{-6}$, and $7.7 \times 10^{-9}$ cm$^2$ V$^{-1}$ s$^{-1}$, respectively. These ambipolar transport properties were attributed to the acridinium cation and fluorenyl anion, which serve as positive and negative charge carriers, respectively. From the above data, a good relationship between electronic effects and hole mobility can be seen. The hole mobility of the compound with an electron-withdrawing group (Br) was higher than that of the compounds with electron-donating groups (Me, OMe, and thienyl). This implies that enhanced intramolecular charge transfer stabilized the twisted conformer and thereby increased hole transport ability. Improved charge carrier mobility obtained via modification with substituents will be a fascinating feature of these semiconducting molecules.

**X-ray structure analysis.** The structures of the Ph-F'As, except for **5f**, were unambiguously determined by single-crystal XRD as shown in Fig. 7 (Supplementary Figs. 55–62, Supplementary Data 5–12, Supplementary Tables 1–21). Crystal structures of both the folded and twisted conformers were obtained for **5a** and **5e**. Surprisingly, both the folded and twisted conformers were present in one unit cell, in ratio 1:1. This is the first examples of BAEs containing both conformers in one crystal. The structure of methanol adduct **6a** was also determined.

There are two well-known ways for overcrowded BAEs to alleviate steric hindrance in the fjord area[38,39]. One way is twisting, and the other is folding. Twisting generates $\omega$. As is shown in Table 1, the two moieties on each side of the central double bond were twisted by $\omega = 40$–57°. In the case of twisted **5e**, $\omega$ was high as 56.7°, showing considerable deformation from the classical planar ethylene structure. Folding generates dihedral angles included two types: (1) within the fluorene moiety (A–C) and the acridane moiety (D–F) and (2) between the fluorene and acridane moieties (ABC–DEF). For the folded structures of **5c** and **5e**, the dihedral angles between D and F were high at 46.06° and 45.49°, respectively. In addition, the folding angle of A–C was typically bigger than that of D–F, implying that folding of the acridane moiety contributed more to alleviating steric hindrance at higher degrees of folding. This also means that the planar fluorene was relatively rigid. By folding of the acridane moiety, the steric hindrance in the fjord region is further alleviated (Supplementary Note 3 for more discussion).

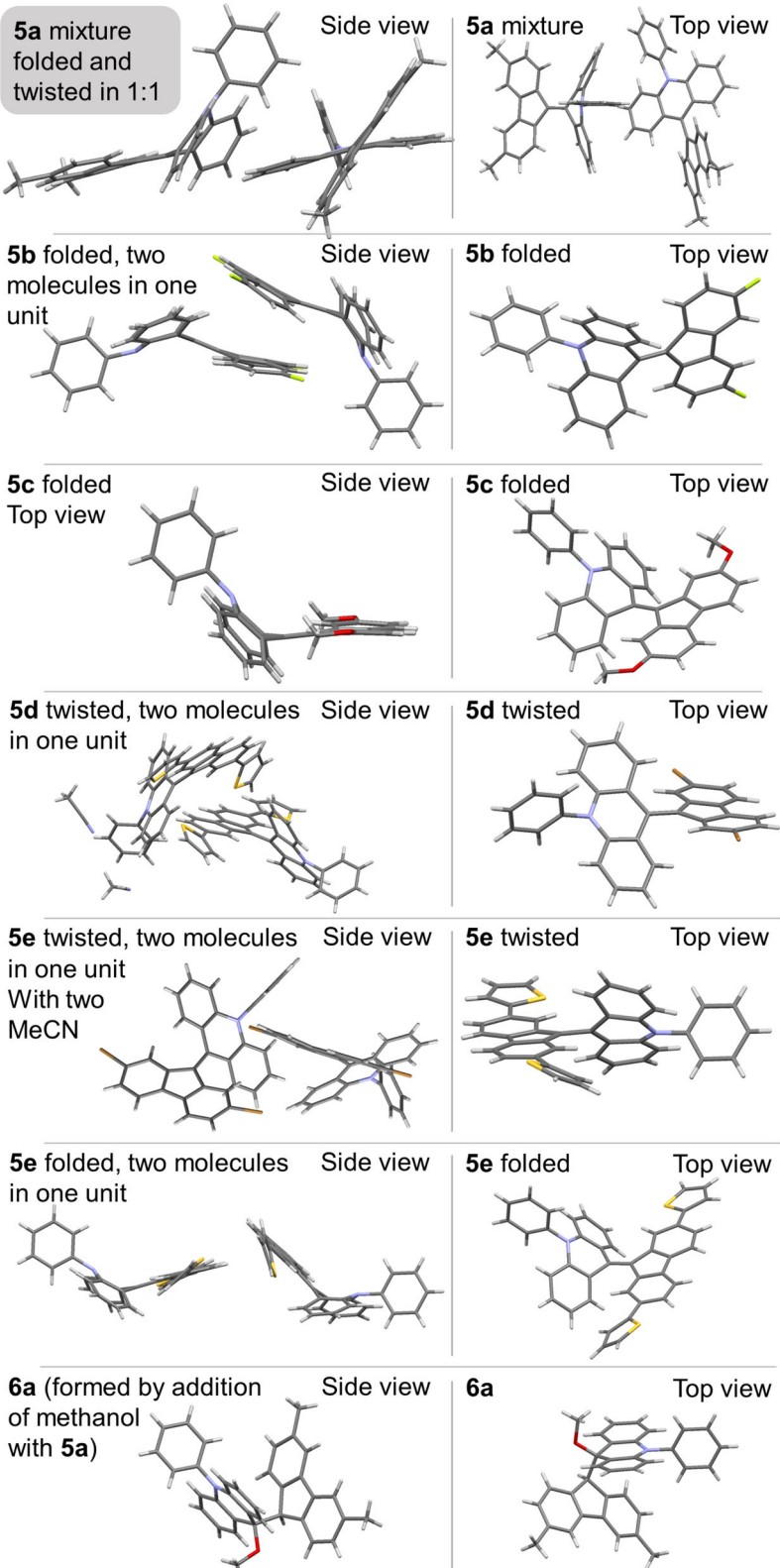

**Fig. 7 Single-crystal X-ray structures.** Structure of **5a** (folded), **5a** (twisted), **5b** (folded), **5c** (folded), **5d** (twisted), **5e** (twisted), **5e** (folded), and **6a**.

For the folded conformers, the bond length between C9 and C9′ ranged from 1.352 to 1.369 Å, slightly longer than the classical C=C bond length (1.34 Å). For the twisted conformers, on the other hand, it ranged from 1.397 to 1.414 Å. We suggest that partial intramolecular charge transfer from the acridane moiety to the fluorene moiety elongated the C9′=C9 bond. Folded **5e** was selected for analyzing the fjord region of Ph-F'As. The nonbonding distances of C1···C1′ (3.000 Å), C8···C8′ (3.055 Å), C1···H1′ (2.318 Å), C8···H8′ (2.216 Å), and H8···H8′ (2.306 Å) in the fjord region imply 14.0, 12.1, 23.8, 4.1, and 24.3%

**Table 1 Selected geometric parameters of Ph-F'As (5a, 5b, 5c, 5d, and 5e) obtained from their crystal structures.**

|  | ω/° | A–C/° | D–F/° | ABC–DEF/° | C9 = C9'/Å | C1···C1'/Å |
|---|---|---|---|---|---|---|
| **5a** folded | 1.77 | 16.58 | 42.52 | 33.91 | 1.359 | 3.060 |
| **5a** twisted | 40.10 | 3.731 | 1.98 | – | 1.397 | 3.015 |
| **5b** folded | 3.08 | 12.75 | 42.88 | 29.73 | 1.361 | 3.005 |
| **5c** folded | 0.00 | 6.46 | 46.06 | 44.75 | 1.352 | 3.033 |
| **5d** twisted | 42.85 | 7.79 | 12.02 | – | 1.414 | 3.068 |
| **5e** folded | 3.45 | 4.69 | 45.49 | 44.72 | 1.369 | 3.000 |
| **5e** twisted | 56.69 | 2.64 | 19.63 | – | 1.402 | 3.092 |

penetration (the van der Waals radii of carbon and hydrogen are 1.71 and 1.15 Å respectively).

The crystallinity of Ph-F'As in this work was generally higher than that of the compounds in our previous work[16,17], presumably because introducing functional groups onto the fluorene moiety led to stronger dipole–dipole interactions between molecules. Furthermore, the crystallinity of the folded conformer was generally higher than that of the twisted conformer, as demonstrated by Ph-F'As powder obtained after solvent evaporation changing from the bluish green twisted conformer to the yellow folded conformer as it dried further. The packing force between the rigid structures of the folded conformers may account for this. The higher crystallinity of the folded conformer compared with the twisted one serves as the foundation of the mechanochromic behavior of Ph-F'As, which have potential applications as smart materials in pressure, heat, and other types of sensors.

## Discussion

We have successfully synthesized Ph-F'As, a new class of BAEs. Introduction a series of substituents onto the fluorene moiety provides a facile way to tune the electronic properties of the Ph-F'As. As we anticipated, directly modifying the π-conjugation systems of FAs enabled more effective control of the equilibrium between their folded and twisted conformers.

Although the equilibrium between BAE conformers has been broadly accepted in previous BAE research, this was the first study to measure the equilibrium constant of a BAE. Compound **5e** had an equilibrium constant of $K = 3.86$ at 30 °C in DMF, with the folded conformer being strongly favored over the twisted conformer. The twisted conformer had a molar absorption coefficient more than six-fold that of the folded conformer, explaining why sometimes the color of the less favored conformer was observed. With increasing temperature, the equilibrium shifted toward the folded conformer in solution. By observing this behavior, we obtained thermodynamic parameters including the changes in enthalpy, entropy, and Gibbs free energy. The change in enthalpy ($\Delta H = 4.21$ kJ mol$^{-1}$) indicated the conformational change from the twisted to folded conformer is endothermic. The change in free energy ($\Delta G = -3.43$ kJ mol$^{-1}$) revealed that the folded conformer is more stable, and the very small change in entropy ($\Delta S = 0.0252$ kJ mol$^{-1}$) confirmed that this change occurs intramolecularly. This in-depth understanding of the thermodynamics of BAEs in equilibrium provides useful information for designing BAEs and related molecular machines for fundamental and applied research.

Controlling the electronic property unexpectedly resulted in obtaining single crystals containing both folded and twisted conformers in one unit cell. This will be an interesting topic of research in crystal engineering, single-crystal electronics, and solid-state physics. To extend this research to doping technology for organic electronics, controlling the folded/twisted conformer ratio to alter this 1:1 ratio in the solid state will be a future research target.

The FAs in this work exhibited various chromic behaviors including mechanochromism in the ground state to achieve color change in response to mechanical stimuli. The yellow folded form could be absorbed into paper and changed to the green twisted form by applying pressure to draw on the paper. This demonstrates the potential application of FAs in reversible surface-pressure sensors. From the perspective of synthetic chemistry, dibromo FA (**5d**) and bis(thienyl) FA (**5e**) could be useful building blocks in polymers and various functional organic materials.

## Methods

**General remarks**. All reagents commercially available were used as received without further purification. $^{1}$H and $^{13}$C NMR spectra were respectively recorded at 500.16 MHz and 125.77 MHz on a JEOL ECZ-500 system. High-resolution (HR) mass spectra were obtained by MALDI using a time-of-flight mass analyzer on a Bruker Ultra exTOF/TOF spectrometer. The UV–Visible spectrum was recorded on a JASCO V-570 spectrometer. CV measurements were performed with a HOKUTO DENKO HZ-5000 voltammetric analyzer.

*General procedures for synthesis of precursors and* **5a–f**: The detailed synthetic process of precursors are included in Supplementary Methods (Supplementary Figs. 1–8, Supplementary Note 1–2). Into a Schlenk bottle, 10-phenylacridine-9 (10H)-thione (**4**, 1 equiv) and triphenyl phosphine (1 equiv) are added, and then put the rubber cover on. The reaction system is designed to keep normal N$_2$ pressure by using a Schlenk line. To start the reaction, anhydrous xylene 4 mL mmol$^{-1}$ was injected into the bottle, and diazofluorene **3** (1 equiv) was added drop-wise during the reaction. The mixture was refluxed totally for 1 h. After removing the solvent by reduced pressure distillation, the mixture was charged on a silica gel short column, and then dichloromethane was passed through the column to remove some impurities. And then, a triethylamine/dichloromethane mixture (1/1 to 1/5) was passed to desorb the product from silica gel, collecting the dark green or blue compounds. Second stage silica gel column chromatography, which was prior treated with triethylamine, was performed using petroleum ether/ dichloromethane (10/1) eluent to obtain the target compounds **5**, yield 35–70%.

*Computational calculation*: All calculations were carried out by using Gaussian09 package at the B3LYP. A 6–31G(d) basis set was used for each level. The calculation levels are described as "B3LYP/ 6–31G(d)".

*Electrochemistry*: Cyclic voltammetry and differential pulse voltammetry were performed using CHI660E voltammetric analyzer. All measurements were carried out in a one-compartment cell under argon gas, equipped with a platinum working electrode, a platinum wire counter electrode, and an Ag/Ag$^+$ reference electrode. The supporting electrolyte was a 0.1 mol L$^{-1}$ dichloro-methane solution of tetrabutylammonium hexafluorophosphate (TBAPF$_6$).

*X-ray single-crystal analysis*: Orange crystals of the folded conformers of **5a**, **5b**, **5c**, and **5e**, dark crystals of **5a** and **5e**, which consist of twisted and folded 1:1, dark crystals of twisted **5d** and **6a**, were obtained from bilayer solution systems using dichloromethane as good solvent and using methanol, acetonitrile, or petroleum ether as poor solvent, or slow evaporation of dichloromethane solution at room temperature. Single-crystal X-ray diffraction data were collected on a diffractometer equipped with a CCD area detector using graphite-monochromated Cu Kα radiation ($\lambda = 1.54184$ Å). Using Olex2, the structure was solved with the ShelXS structure solution program using direct methods and refined with the ShelXL refinement package using least squares minimization.

*SCLC measurements*: The mobility was determined by fitting the dark current to a model of a single-carrier SCLC, which is described by the equation: $J = \frac{9}{8}\varepsilon_0\varepsilon_r\mu\frac{V^2}{L^3}$, where $J$ is the current density, $\mu$ is the mobility, $\varepsilon_0$ is the permittivity of free space, $\varepsilon_r$ is the relative permittivity of the material, $L$ is the thickness of the Ph-F'As compound layer, and $V$ is the effective voltage. The thickness of the thin films was measured with a DEKTAK 6 M stylus profilometer. A solution of the Ph-F'As compound was spin-coated onto the Al/glass to form a thin film in electron-only devices, and onto the PEDOT:PSS in hole-only devices. The LiF/Al electrodes (LiF = 0.6 nm; Al = 100 nm) or MoO$_3$/Al were evaporated onto the Ph-F'As compound thin films in the electron-only devices and hole-only devices, respectively. The experimental dark current density $J$ was measured under an applied voltage swept from –5 to 5 V.

## Data availability

The authors declare that the data supporting the findings of this study are available within the paper and its supplementary information files, or from the corresponding author upon reasonable request. CCDC contains the supplementary crystallographic data for this paper. CCDC numbers for folded **5a**, both folded and twisted **5a**, folded **5b**, folded **5c**, twisted **5d**, folded **5e**, twisted **5e**, and **6a** are 1985971, 1985972, 1985973, 1985983, 1985984, 1985985, 1985986, and 1985989. These data can be obtained free of charge from The Cambridge Crystallographic Data Centre via www.ccdc.cam.ac.uk/ data_request/cif.

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

## Acknowledgements

We appreciate funding from University of Science and Technology of China. This work was also supported by Grants-in-Aid for Scientific Research (JSPS KAKENHI Grant Number JP19K22181) from the Ministry of Education, Culture, Sports, Science and Technology (MEXT), Japan. The computations were performed using Research Center for Computational Science, Okazaki, Japan.

## Author contributions

Y. Matsuo and Y. Wang conceived and designed the project and wrote the paper. Y. Ma contributed to analysis of thermodynamic studies. K. Ogumi, B. Wang, and Y. Fu carried out computational calculations. T. Nakagawa measured the SCLC mobility.

## Competing interests

The authors declare no competing interests.
