## [Peer Review File · Communications Chemistry]

Reviewers' comments:

Reviewer #1 (Remarks to the Author):

Please find the attached word file for my review comments.

Reviewer #2 (Remarks to the Author):

In this manuscript, Yutaka Matsuo and co-workers have reported a series of new organic compounds derived from N-phenyl-substituted fluorenylidene-acridane. The authors determined the equilibrium constants, changes in enthalpy, entropy, and free energy of the compounds in solution to give an in-depth understanding of the equilibrium behaviors of overcrowded alkenes. They also demonstrated the mechanochromism, thermochromism, solvatochromism, vapochromism and proton-induced chromism properties as well as the electrochemistry and charge carrier transport performance of the dyes. Concurrently, possible mechanisms for their chromism phenomena were proposed. Although some experimental results from this work are interesting, in my opinion, it can not hit the high-quality requirement of Communications Chemistry. Consequently, it is not sufficient to be published on the Journal at this stage. Some of my comments are listed as follows:

1. Sample purities of compounds 5a-e are not sufficient because peaks from impurities can be clearly observed in their ¹H NMR spectra, especially in the range of 5.0 to 9.5 ppm. This is a critical error. In this context, most of the experimental results are likely to be unreliable.
2. As demonstrated by the authors, 5a contained both the twisted and folded conformations in one single unit cell, and the molar absorption coefficient of the twisted conformer was larger than the folded one. Why did the compound show yellow rather than green color in the crystalline state?
3. Chromism properties were found from the compounds in the solid state. However, I can't find any solid-state UV-visible absorption spectra for the dyes. Moreover, only chromism data of 5a and 5c were mentioned in the manuscript. How about the other compounds? Details on the photophysical properties of all the luminogens are suggested to be provided and necessary discussion should be given.
4. The authors claimed that the transformation from twisted to folded conformer is an endothermic process, which was certified by the positive value of ΔH . But, in the study of mechanochromism of 5a, they stated that the exothermic transition was responsible for the transformation of the twisted conformation to the folded one, leading to the recovery of color from green to yellow upon heating. Rational explanation should be given for this contradictory result. In fact, thermal transitions on the DSC curves in Figure 3c and Figure S42 are not clearly enough.
5. The first heating runs for the DSC curves of the samples are important for elucidating the possible mechanochromism mechanism of the compounds. Consequently, it is not appropriate to omit them. On top of this, XRD patterns and DSC curves for the pristine, ground, fumed and annealed samples are suggested to be supplied to give a deep insight into their mechanochromism.
6. 5c was found to exhibited proton-induced chromism. Why did authors use 5a to explained its chromism mechanism? Furthermore, evidence should be offered to support this proposed mechanism.
7. Opposite results were achieved on energy difference of the twisted and folded conformations by using M062X and B3LYP for theoretical calculations. What is the final conclusion on this issue? How about the results of compounds 5b, 5c, 5d and 5f?

Reviewer #3 (Remarks to the Author):

This manuscript reports the functionalization of the fluorene moiety of Fluorenylidene-Acridane and the equilibrium of the folded and twisted conformers, which is an ongoing work of the author's

previous reports. The various chromisms upon external stimuli are expectable. But this is the first time that the authors discuss the determination of the equilibration of the two conformers. The conformational isomerization was investigated by theoretical calculations and some key structures were confirmed by crystallographic analysis. However, there are still some issues that prohibit me from supporting this manuscript as listed below, especially the first one.

1) the authors should distinguish the difference between copper-catalyzed Ullmann amination and the palladium-catalyzed Buchwald-Hartwig amination, as depicted in the first reaction of Scheme 1 page and the bottom 2 lines in page 5.

2) In the reference part, ref 6, the author's own paper, the final page is obviously not right; ref 8 and 27c, the paper titles are missing; some paper titles involve capitalized letters, some are not, which should be unified; a space is missing (Moore, J, S.) in ref 7

In this paper, Matsuo and coworkers demonstrate mechanochromic behavior of new BAE compounds which are composed of acridane and fluorenylidene. Key point of this research is the development of new synthetic methodology, as shown in Scheme 1, which allows to synthesize many derivatives with a variety of substituent groups on the fluorenyl ring. Thanks to the efficient synthetic protocol, the electronic structure of the new BAEs can be tuned precisely, leading to detailed investigation on the mechanism of conformational change upon external stimuli. Another key feature of this paper is various chromic behavior of the new BAEs and their high mechanical stability may be of great interest to the readership, especially chemists of functional dyes. In view of a high quality of the study, I think that this paper is publishable in Communications Chemistry. However, on this occasion, the authors have a chance to revise the paper. My minor comments are as follows:

1. Page 3, reference “5,6” appears after “7-10” and “11-16”. Correct it.
2. Page 4, ab initio and DFT are different calculation theories.
3. Page 7, **5a** adopts both folded and twisted forms in crystal. Do DFT calculations support small energy difference between both forms for **5a**? Do the other BAEs (**5b-5f**) show larger energy differences?
4. Page 11, the authors assume that the vapochromism of **5a** is related to micro-crystallization induced by solvent vapor. This assumption would be confirmed by powder XRD measurements.
5. Page 13, the authors describe that the red-shift of **5f** is attributable to increased charge transfer degree. Is it possible to estimate charge transfer degree by DFT calculations? The charge density of the nitrogen atom of **5a**, **5c** and **5f** might be related to their charge transfer degrees.
6. Page 15, what is the evidence for the concerted mechanism for the formation of the methanol adduct **6a**?
7. Figure 5a, the authors assume that the protonation occurs at the carbon atom of the fluorenyl ring. Why doesn't protonation occur on the nitrogen atom?

8. Page 17, what does SI indicate? Figure S59?

9. Page 17, why did the authors use B3LYP/6-31G(d) instead of M06/6-31G(d)? The latter method is much better for the calculation of sterically congested molecules.

- Key results: described above.
- Validity: the manuscript have no flaw.
- Originality and significance: the results are of great interest to organic chemists and material scientists working on functional dyes.
- Data & methodology: some DFT calculations should be added, as indicated above.
- Appropriate use of statistics and treatment of uncertainties: curve fitting parameters in Figure S63 are indicated in Japanese.
- Conclusions: the conclusion is robust, valid and reliable.
- Suggested improvements: described as minor comments.
- References: this manuscript reference previous literature appropriately.
- Clarity and context: Abstract, introduction and conclusions are appropriate.

Comments from the reviewer #1

1. Page 3, reference “5,6” appears after “7-10” and “11-16”. Correct it.

(response) We corrected this mistake.

2. Page 4, ab initio and DFT are different calculation theories.

(response) This referred work (ref 17) is relatively old and did calculation with neither ab initio nor DFT. Also, this sentence mentions very general thing and now we found this sentence is not so meaningful. So, we delete this sentence to make concise our paper. Instead, we added experimental knowledge with a historically important paper.

Small energy difference between the two conformers of BAEs have been experimentally known [17]. [17] W. Theilacker, G. Kortüm, G. Friedheim, Chem. Ber. 1950, 83, 508–519.

3. Page 7, **5a** adopts both folded and twisted forms in crystal. Do DFT calculations support small energy difference between both forms for **5a**? Do the other BAEs (**5b-5f**) show larger energy differences? Do the other larger energy differences?

(response) Because UV-Vis spectra pattern are similar between **5a-f**, we consider all these compounds have small energy difference between folded and twisted conformers. We typically did calculation for **5e** with the M06 method. We think **5a** is in the similar situation to **5e** because of electron-donating groups on the fluorene parts. So, both compounds have small energy differences, being accordance with folded and twisted forms in the crystal.

4. Page 11, the authors assume that the vapochromism of **5a** is related to micro-crystallization induced by solvent vapor. This assumption would be confirmed by powder XRD measurements.

(response) We appreciate the reviewer's suggestion. First, we have already showed and discussed powder XRD data after grinding and after fuming for our previous compound in our previous paper (ref. 15). We performed powder XRD measurement of **5a** after grinding and after fuming, and added this data in SI (Powder XRD, figure S74). After grinding yellow **5a**, ground green **5a** was obtained. Powder XRD measurements of ground **5a** (before fuming) and **5a** after fuming with dichloromethane (DCM) vapor were carried out. By comparison, after fuming, many diffraction peaks appeared with high intensities in a range from $2\theta = 10^\circ$ to 30° , indicating the process of micro-crystallization.

Figure S74. Powder XRD measurements of **5a** before and after fuming with DCM vapor.

5. Page 13, the authors describe that the red-shift of **5f** is attributable to increased charge transfer degree. Is it possible to estimate charge transfer degree by DFT calculations? The charge density of the nitrogen atom of **5a**, **5c** and **5f** might be related to their charge transfer degrees.

(response) We appreciate the reviewer for suggesting potentially nice method for evaluating charge transfer degree. Calculation of the charge transfer density of the nitrogen atom would work well but be challenging, we would like to consider this method in our future work.

6. Page 15, what is the evidence for the concerted mechanism for the formation of the methanol adduct **6a**?

(response) Many thanks for your question. We also considered protonation to the fluorene part occurs first. However, as we know that pKa of MeOH is 15.5, MeOH is not easy to dissociate to H⁺ and MeO⁻. In other words, concentration of H⁺ and MeO⁻ in the reaction system is low. It might be different from the protonation experiment with acetic acid shown in Figure 4d. Thus, we propose a concerted process containing both nucleophilic (protonation) and electrophilic processes. This is our speculation. So, we modified sentences as follows.

"Formation of the methanol adduct **6a** proceeded via a concerted mechanism as shown in Figure 5b."  "Formation of the methanol adduct **6a** proceeded as shown in Figure 5b. Considering pKa of methanol is 15.5, we assume the reaction occurs via a concerted mechanism."

7. Figure 5a, the authors assume that the protonation occurs at the carbon atom of the fluorenyl ring. Why doesn't protonation occur on the nitrogen atom?

(response) We thank this confirmation. First, as we know the basicity of triphenylamine is much weaker than weak base trimethylamine. The pKa of trimethylamine is 2×10^{16} times larger than the pKa of triphenylamine. From the bond-energy databank, <http://ibond.chem.tsinghua.edu.cn>, we know that pKa of trimethylamine is =17.61 in acetonitrile,¹ pKa of triphenylamine is 1.3 in acetonitrile.² That means the nitrogen atom in the acridine part has almost no basicity. This is because the p- π conjugation with phenyl rings and lone pair electrons of the nitrogen atom will lead almost no basicity. The strongest evidence of the protonation at the fluorene part is isolation of methanol adduct that has the hydrogen atom on the fluorene moiety.

Second, considering the fact that acridinium structure is very stable and common structure,³⁻⁶ the protonation must occur at the fluorene part to form the acridinium structure.

Reference:

- 1 *European Journal of Organic Chemistry*, **2012**, 2167–2172.
- 2 *Journal of the American Chemical Society*, **1965**, 87 (22), 5005–5010.
- 3 *Egyptian Journal of Analytical Chemistry*, (1994), 3(1), 99–102.
- 4 *Organische Chemie*, (1984), 39B(10), 1399–408.
- 5 *Photochem*, (1978), 7, 349–54.
- 6 *Journal of biochemical and biophysical methods*. (2002), 53(1–3), 1–14.

Comments from the reviewer #2

1. Sample purities of compounds 5a-e are not sufficient because peaks from impurities can be clearly observed in their ¹H NMR spectra, especially in the range of 5.0 to 9.5 ppm. This is a critical error. In this context, most of the experimental results are likely to be unreliable.

(response) We appreciate the reviewer giving this comment. We think this is the most important criticism from this reviewer. At this point, however, we need to explain this is typical observation in the NMR of bistricyclic aromatic enes (BAEs) that have two **conformational** isomers. In the NMR time scale, it is not like rapid interconversion, but we can observe one major peaks set of one isomer and another minor peaks set of the other isomer. In addition, for some compounds with different isomerization speed, peaks become broad due to interconversion between two conformers. For example, in Figure S25 in SI for the ¹H NMR of **5b**, there exists a minor peaks set around 6.73 ppm and 8.39 ppm, which are from a minor conformer (folded or twisted; likely twisted judged from UV-Vis studies). These are not impurity peaks, but peaks from the minor conformer.

These compounds **5a-f** can be easily purified by silica gel column chromatography because polarity of these compounds are quite different from the starting materials and side products due to their polarized structures (δ^+ in the acridane part, δ^- in the fluorene part), and high-resolution MS data for **5a-f** were matched very much, and also we obtained almost all single crystal structures for **5a-e**, except for **5f**; Compounds with impurities are normally not easy to form single crystals in good quality.

Most importantly, there are many literatures showing small peaks of the minor isomer involving the isomerization process. We would like to show an organized figure taken from literatures and references below.

Figure. ¹H NMR spectra of BAEs in conformational isomerism. (a) Isomerization progress of *syn*-*anti* isomer in CDCl₃ in ref. R3. (b) Progress *E,Z*-diastereomerization as a function of time in ref. R5. (c) Progress of *syn*-to-*anti* in ref. R6.

Reference

- R1. *Eur. J. Org. Chem.* **2001**, 15–34.
 R2. *J. Am. Chem. Soc.* **2003**, *125*, 12829–12835.
 R3. *Chem. Sci.* **2011**, *2*, 2029.
 R4. *Asian J. Org. Chem.* **2015**, *4*, 1392–1398.
 R5. *Chirality.* **2015**, *27*, 919–928.
 R6. *Structural Chemistry.* **1993**, Vol. 4, No. 1.

2. As demonstrated by the authors, 5a contained both the twisted and folded conformations in one single unit cell, and the molar absorption coefficient of the twisted conformer was larger than the folded one. Why did the compound show yellow rather than green color in the crystalline state?

(response) We thank this reviewer asking this question. We can answer to this question easily. The crystals in Figure S48 contain the folded conformer only having greenish yellow color. The crystals in Figure S49 contain both folded and twisted conformers showing black color.

The folded conformer (greenish yellow) in **Figure S48**.

The mixture of the folded and twisted conformers (black) in **Figure S49**.

3. Chromism properties were found from the compounds in the solid state. However, I can't find any solid-state UV-visible absorption spectra for the dyes. Moreover, only chromism data of **5a** and **5c** were mentioned in the manuscript. How about the other compounds? Details on the photophysical properties of all the luminogens are suggested to be provided and necessary discussion should be given.

(response) Solid state UV-Vis absorption spectra of the yellow (folded) form, the green (twisted) form after grinding, and the yellow (folded) form after fuming have been already reported in our previous paper (ref. 15). Because the main topic of this work is equilibrium and thermodynamics, we think we should not repeat similar discussion to our previous paper. We have shown solid-state UV-Vis absorption of **5a** in the overlaid graph of the Figure 3d. We kept this figure and improved Figure 3 caption for the clarity.

Regarding various chromic behavior other than **5a**, again we are thinking we should present chromic part as simple as we can (because of the same reason; we should not repeat similar discussion to our previous discussion to focus on our main topic in this paper). However, we understand reviewer's suggestion, we added several figures in SI. Compounds **5b-f** show similar chromic behavior to **5a**.

4. The authors claimed that the transformation from twisted to folded conformer is an endothermic process, which was certified by the positive value of ΔH . But, in the study of mechanochromism of **5a**, they stated that the exothermic transition was responsible for the transformation of the twisted conformation to the folded one, leading to the recovery of color from green to yellow upon heating. Rational explanation should be given for this contradictory result. In fact, thermal transitions on the DSC curves in Figure 3c and Figure S42 are not clearly enough.

(response) We appreciate the reviewer's good question. We can say equilibrium in the solution

state and morphology change in the solid state are not related each other in this case. Transformation from the twisted to folded conformers in the solution state is endothermic, which was certified by the positive value of ΔH . On the other hand, thermal ordering from the twisted to the folded forms in the solid state occurs by heating around at 70 °C with exothermic process (but not obvious exothermic peaks in DSC, it shows glass transition) due to intermolecular interaction such as packing force in the folded crystal, like fuming-induced crystallization into the folded one. We added this explanation into the main text. We thank this reviewer to improve this manuscript.

Original: This behavior can be explained by recrystallization to the folder conformer over the glass transition temperature (around 70 °C). Differential scanning calorimetry (DSC) measurement of **5a** supports this, showing a broad exothermic peak around this temperature. For other Ph-F'As, this exothermic peak ranged from 65 °C to 75 °C.

Revised: This behavior can be explained by recrystallization to the folder conformer over the glass transition temperature (around 70 °C). Differential scanning calorimetry (DSC) measurement of **5a** supports this, showing a broad **and small** exothermic peak around this temperature. For other Ph-F'As, this exothermic peak ranged from 65 °C to 75 °C. **Transformation from the twisted to folded conformers in the solution state was endothermic, which was certified by the positive value of ΔH . On the other hand, thermal ordering from the twisted to the folded forms in the solid state occurs by heating around at 70 °C with exothermic process because of intermolecular interaction such as packing force in the folded crystal, like fuming-induced crystallization into the folded one.**

Regarding clearness of the DSC curves, we revised them in Figure S42 as shown below. We enlarged letters in this figure to see clearly. We appreciate this reviewer to improve this paper.

Figure S42. Differential scanning calorimetry (DSC) data of FAs. (a) **5a**, (b) **5b** (c) **5c**, (d) **5d**, and (e) **5e**. The first scan of the heating and cooling processes are shown. The sharp endothermic peak found in **5c** indicates melting process. Glass transitions were observed around 65–75 °C.

5. The first heating runs for the DSC curves of the samples are important for elucidating the possible mechanochromism mechanism of the compounds. Consequently, them. On top of this, XRD patterns and DSC curves for the pristine, ground, fumed and annealed samples are suggested to be supplied

to give a deep insight into their mechanochromism.

(response) The reviewer is right. We checked our data and found it was the first scan. We updated Figure S42 for DSC curve as described above.

As for the XRD patterns, we measured powder XRD of **5a** after grinding (before fuming) and after fuming. We added this figure to SI (Figure S74). In addition, we measured powder XRD of **5a** before and after grinding of the pristine folded powder. It was also added to SI (Figure S75). These results are accordance with chromic observation and DSC data.

Figure S74. Powder XRD measurements of **5a** before and after fuming with DCM vapor.

Figure S75. Powder XRD measurements of **5a** before and after grinding upon pristine folded state.

6. **5c** was found to exhibit proton-induced chromism. Why did authors use **5a** to explain its chromism mechanism? Furthermore, evidence should be offered to support this proposed mechanism.

(response) All compounds **5a-f** can be protonated and show proton-induced chromism. First we corrected our mistake of **5c** into **5a** in both Figure 4d and the main text. Because the protonation study is remote from the main important topic (equilibrium and thermodynamics) of this work, we only

show the protonation of **5a** in Figure 4d as a typical example with a simple substituent (methyl group) not like thiophene (can be protonated) rings in **5e**. Finally, we succeeded in the single-crystal X-ray analysis of the methanol-adduct (**6a**) with simple methyl substituents. This is the biggest evidence for protonation occurring at the fluorene part.

When we consider pK_a of methanol (15.5), we can think methanol is not easy to dissociate to H^+ and MeO^- . The methanol addition (Figure 5b) will be different from the protonation (Figure 5a) with acetic acid. Thus, we propose a concerted process containing both nucleophilic (protonation) and electrophilic processes. We modified sentences to make explanation clearer.

"Formation of the methanol adduct **6a** proceeded via a concerted mechanism as shown in Figure 5b."  "Formation of the methanol adduct **6a** proceeded as shown in Figure 5b. **Considering pK_a of methanol is 15.5**, we assume the reaction occurs via a concerted mechanism."

7. Opposite results were achieved on energy difference of the twisted and folded conformations by using M062X and B3LYP for theoretical calculations. What is the final conclusion on this issue? How about the results of compounds **5b**, **5c**, **5d** and **5f**?

(response) We thank the reviewer pointing out this question. The final conclusion is **energy of the folded and twisted conformers are very similar**. The different result is because the M06 method can estimate weak intermolecular interaction; and the B3LYP method cannot include intermolecular interaction. We kept the M06 data only in the main text because intermolecular interaction is important in conformational isomerism. So, we moved less important B3LYP data into the SI.

We conducted the M06 calculation for only **5e**, which is the most important compound that we used in equilibrium and thermodynamics studies (heart of this work). Also, the single crystals of **5e** contain both the folded and twisted conformers, which implies energies of the folded and twisted conformers are very similar. Other derivatives (**5a**, **5c-f**) are less important than **5e**. This is demonstration of derivatization, and there is a meaning that we found a good-balanced compound (**5e**) by versatile derivatization. We consider all compounds **5a-f** have similar energies in both folded and twisted conformation, which can be judged from UV-Vis spectra with similar folded/twisted ratio.

We found the HOMO/LUMO calculation part overlapped in the main text and SI. We deleted this part from the main text. We keep it in SI.

Comments from the reviewer #3

1) the authors should distinguish the difference between copper-catalyzed Ullmann amination and the palladium-catalyzed Buchwald-Hartwig amination, as depicted in the first reaction of Scheme 1 page and the bottom 2 lines in page 5.

(response) We used copper for this amination. We thus corrected the name. We thank this reviewer to tell us this.

2) In the reference part, ref 6, the author's own paper, the final page is obviously not right; ref 8 and 27c, the paper titles are missing; some paper titles involve capitalized letters, some are not, which should be unified; a space is missing (Moore, J, S.)in ref 7

(response) We corrected mistakes in the reference section very carefully. We appreciate this reviewer.

Reviewers' comments:

Reviewer #1 (Remarks to the Author):

I reviewed the revised paper. What I pointed out in the original paper are properly answered.

Reviewer #2 (Remarks to the Author):

The authors have almost addressed all the comments. I would like to recommend its publication after minor revising the manuscript according to the following two issues.

1. For the ^1H NMR spectra, they argued that the peaks are attributable to minor conformers of the compounds of interest. However, there are additional peaks in the spectrum of compound 5b in particular which do not seem to fit this hypothesis (e.g. at $\delta = 4.4, 4.1, 3.5,$ and $1.5\text{-}0.8$ ppm). The authors should further specify the reason. In addition, their reply to my comments should be added into the main manuscript to clarify these issues to readers.
2. The thermal transitions on the DSC curves in Figure S42 remain difficult to identify. It is suggested to provide larger diagrams and remove the cooling curves to make the peaks clearer.

Responses

Reviewer #2:

The authors have almost addressed all the comments. I would like to recommend its publication after minor revising the manuscript according to the following two issues.

1. For the ^1H NMR spectra, they argued that the peaks are attributable to minor conformers of the compounds of interest. However, there are additional peaks in the spectrum of compound **5b** in particular which do not seem to fit this hypothesis (e.g. at $\delta = 4.4, 4.1, 3.5,$ and $1.5\text{-}0.8$ ppm). The authors should further specify the reason. In addition, their reply to my comments should be added into the main manuscript to clarify these issues to readers.

Thanks for your question. There were residual solvents peaks retained in **5b**, petroleum ether (0.8 (m) and 1.2 (m) ppm), ethyl acetate (1.1 (t), 2.0 (s), and 4.0 (q) ppm), and so on. We took ^1H NMR again with single crystals samples in $\text{DMSO-}d_6$ to obtain a new spectra as below.

Figure S25. ^1H NMR (400 MHz) of **5b** in $\text{DMSO-}d_6$.

In addition, their reply to my comments should be added into the main manuscript to clarify these issues to readers.

We appreciate the reviewer giving this comment. We added explanation into the main manuscript for readers. We added one sentence, "Note that ^1H NMR spectra exhibit one major peaks set of one isomer and another minor peaks set of the other isomer.^{2a,19}" and references 19.

2. The thermal transitions on the DSC curves in Figure S42 remain difficult to identify. It is suggested to provide larger diagrams and remove the cooling curves to make the peaks clearer.

Thanks for your suggestion. we modified figure S42 as below, with curves being stacked together, and magnified.

Figure S42. Differential scanning calorimetry (DSC) data of FAs. Brown, gray, blue, red, and green lines stand for DSC curves of **5a**, **5b**, **5c**, **5d**, and **5e**, respectively. The first scan of the heating and cooling processes are shown. The sharp endothermic peak found in **5c** indicates melting process. Glass transitions were observed around 62–74 °C.

REVIEWERS' COMMENTS:

Reviewer #2 (Remarks to the Author):

The authors have addressed all my comments. I would like to recommend it for publication on the journal.